# Mediation role of perceived social support between recurrence risk perception and health behaviour among patients with stroke in China: a cross-sectional study

Xiaoxuan Wang  , Zhen-Xiang Zhang, Bei-Lei Lin, Hu Jiang  , Wenna Wang, Yong-Xia Mei  , Chunhui Zhang, Qiushi Zhang, Su-Yan Chen

Nursing and Health school, Zhengzhou University, Zhengzhou, China

**Correspondence to**
Professor Zhen-Xiang Zhang;
zhangzx6666@126.com and
Dr Bei-Lei Lin;
linda870926@126.com

## ABSTRACT

**Objectives** To examine whether patients who had a stroke with high recurrence risk perception would have healthier behaviour and to explore whether perceived social support would function as a mediator.
**Design** A cross-sectional study.
**Setting** The study was conducted in a public tertiary hospital in China.
**Participants** A total of 254 patients with stroke were invited to participate, and 250 patients with stroke completed questionnaires validly.
**Primary and secondary outcome measures** Questionnaires were administered offline to collect data, consisting of four parts: general demographics and scales related to recurrence risk perception, perceived social support, and health behaviour. A path analysis and correlation analysis were used to analyse the data.
**Results** Out of 250 patients with stroke, 78.4% had moderately low health behaviour. The majority (70.8%) of these patients were elderly. High recurrence risk perception and high perceived social support were significantly associated with better health behaviour (all p<0.001). Perceived social support mediated the relationship between recurrence risk perception and health behaviour after controlling for age, gender, education and monthly income in the regression model (95% CI 0.263 to 0.460) and the effect value was 0.360. It was also confirmed that perceived social support had the highest mediation effect with a proportion of mediation up to 59.31%.
**Conclusions** Recurrence risk perception and perceived social support were influential factors in promoting health behaviour. Moreover, the impact of recurrence risk perception on health behaviour was partially mediated by perceived social support. Therefore, to enhance the sustainability of health behaviour, it is crucial to inform patients with stroke about the risk of recurrence. Patients with more perception of recurrence risk can improve their recovery confidence and thus perceive more social support.

## STRENGTHS AND LIMITATIONS OF THIS STUDY

⇒ A path analysis is used to analyse the associations among perceived social support, recurrence risk perception and health behaviour.
⇒ This study is a cross-sectional survey, and therefore, cannot make causal inferences.
⇒ Survey data from only one region may limit the generalisability of our findings.

## INTRODUCTION

Globally, stroke recurrence is a significant but preventable risk. A study that contained 500 000 Chinese adults showed that the cumulative recurrence rate for patients who had a stroke was 17% at 1 year, 41% at 5 years and 53% at 9 years.[1] Despite the implementation of policies and guidelines in various countries aimed at preventing secondary strokes, there has been no significant decrease in the rates of stroke recurrence.[2–4] The occurrence of stroke is closely influenced by a variety of factors, including behavioural, psychological and social-environmental factors. Among these factors, behavioural risk factors play a significant role in the increasing prevalence of chronic diseases.[5]

Adopting healthy lifestyle habits can significantly reduce the risk of stroke recurrence. Specifically, engaging in regular physical activity, following a balanced diet, quitting smoking and consuming alcohol in moderation have been shown to collectively prevent approximately 80% of stroke recurrences.[6] Despite the proven benefits of healthy behaviour adoption, the actual rate of implementation remains discouragingly low. Liu *et al*[7] investigated 515 patients with stroke and found that the health behaviour level of patients was moderate. In another cross-sectional study[8] involving 112 patients who had a stroke, it was found that only 17% of the participants were adhering to medication



regimens. This highlighted the importance of identifying the factors that influence the health behaviour of patients who had a stroke and investigating strategies to improve their adherence. Previous studies have shown that various factors, including gender,[9] age,[10] level of knowledge,[11 12] self-efficacy,[12 13] recurrence risk perception[14] and social support,[12 15] play a significant role in influencing the health behaviour of patients who had a stroke.

Recurrence risk perception represents a critical factor influencing health behaviour.[14] Recurrence risk perception refers to the perception of early warning features of recurrence, severity, behaviour-related risk factors and disease-related risk factors.[16] The study concluded that stroke patients' perception of recurrence risk was an important predictor for patients to adopt healthy behaviours to control and reduce the recurrence of stroke.[17] However, the precise relationship between risk perception of recurrence and health behaviour remains uncertain. According to the health belief model (HBM),[18] individuals who hold the belief that engaging in risky behaviours carries significant health risks are more likely to adopt healthier behaviours. In other words, recurrence risk perception affects health behaviour positively. On the contrary, a cross-sectional comparative study conducted in China showed that patients with stroke who perceived a higher risk of recurrence were more likely to adopt healthy behaviour.[14] Iversen *et al*[19] investigated 435 patients with stroke and confirmed that a higher seriousness perception of stroke was associated with more help-seeking behaviour and shorter prehospital delay. There is also another argument that recurrence risk perception is detrimental to health behaviour. Freeman-Gibb *et al*[20] recruited 117 patients and found patients who perceive a high risk of recurrence always experience higher levels of anxiety and fear. Excessive anxiety and fear were actually detrimental to the patient's adoption of health behaviour.[21] Consequently, in order to understand how stroke risk perception affects the likelihood of people adopting healthy behaviours, more research is needed.

Social support is another significant impact factor on health behaviour.[22] Social support includes both subjective and objective aspects, and the subjective aspect refers to the perception of social support. It is known that social support is an interactive process that involves providing and receiving social support, which means social support must be perceived by patients so that it can be effective.[23] Besides, a systematic review[24] discovered that the associations between perceived social support (PSS) and health-related quality of life appear to be more frequently significant than the relationships between specific categories or sources of social support and health-related quality of life. Meanwhile, studies[12 15] have indicated that individuals who perceive encouragement, understanding and assistance from their social circles are more likely to adopt and sustain healthy behaviours such as exercise, proper diet and adherence to medical treatments.

PSS refers to an individual's belief in the availability of various social resources, including emotional support,

tangible aid and informational assistance.[25] The Common Sense Model of Self-regulation (CSM) is a theoretical framework that is widely used in the study of illness perception.[26] It was designed to describe dynamic interactions among the variables controlling health behaviour in response to current or future health threats.[27] The CSM comprises six main components[28]: (1) situational stimuli, (2) cognitive illness representations, (3) emotional illness representations, (4) coping strategies, (5) illness and emotional outcomes and (6) coping appraisal. Previous studies have already applied the CSM model to predict health outcomes in diabetes,[29] explore the experience of breathlessness[30] and predict healthy eating among individuals at risk of metabolic syndrome.[29] For patients with stroke, the surrounding situational stimuli (including endogenous or social factors) cause patients with stroke to develop cognitive and emotional illness representation (recurrence risk perception). Cognitive and emotional illness representation are also continually updated[28] in response to memories of new information (eg, PSS), which affects coping strategies (the choice of adopting health behaviour). Therefore, it is reasonable to choose the CSM model as the theoretical framework for our study.

According to the CSM, we speculated the effect of recurrence risk perception on health behaviour was partly mediated by PSS. Nevertheless, there is a significant gap in the knowledge regarding the impact of an individual's perception of stroke recurrence risk on their sense of social support and subsequent emotional consequences.[31] In fact, in traditional Chinese philosophy, the concepts of 'recurrence' and 'risk' are often met with fear and apprehension. Patients who had a stroke may experience avoidance or isolation behaviour as a result of their fear, subsequently resulting in a decline in their PSS networks,[32] ultimately reducing the perceived level of social support for patients who had a stroke. Therefore, we hypothesised that recurrence risk perception will be negatively and significantly related to PSS. Moreover, patients with higher recurrence risk perception may be prevented from adopting healthy behaviour due to negative emotions. It proposes the following hypothesis (figure 1). This study aimed to examine the relationship between the recurrence risk perception and health behaviour among patients with stroke and explore the role of PSS in this relationship.

Hypothesis 1: Recurrence risk perception is negatively correlated with health behaviour.

Hypothesis 2: Recurrence risk perception is negatively associated with PSS.

Hypothesis 3: PSS positively and significantly relates to health behaviour.

Hypothesis 4: PSS partially mediates the relationships between recurrence risk perception and health behaviour.

## METHODS
### Participants and procedure
Convenience sampling was used to conduct this cross-sectional survey from June to August 2021. Inpatient

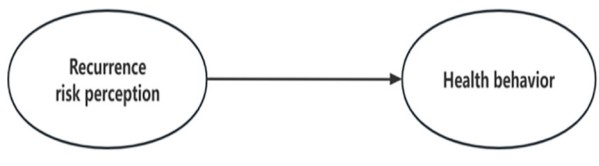

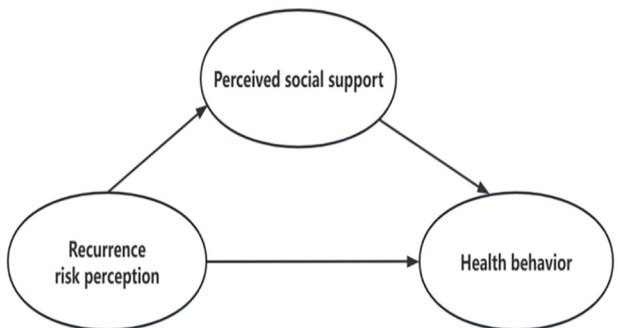

**Figure 1** The hypothesis model of recurrence risk perception, health behaviour and perceived social support in patients with stroke.

patients with stroke were selected from the department of neurology at a tertiary hospital in Luoyang City, Henan Province, China. The following were the inclusion criteria: (a) patients were diagnosed with stroke by MRI or CT[33]; (b) during the recovery stage (2 weeks to 6 months after onset[34]); (c) having the ability to communicate with researchers verbally; (d) over 18 years. Exclusion criteria: (a) combined with severe cardiac, liver and renal dysfunction or other malignant tumours and (b) a history of serious mental illness or a family history of serious mental illness. Ping[35] proposed that the sample size should be at least 5–10 times the independent variables. There were 16 variables in this study (13 general information questions, the recurrence risk perception of stroke scale, the PSS scale (PSSS) and the Health Behaviour Scale for patients with stroke (HBS-SP)). Considering 20% of invalid questionnaires, the minimum sample size of the study was 200 patients with stroke; hence, a sample size of 250 met the requirement for testing the hypothesised models.

### Data collection

To ensure the validity and reliability of our study, we first conducted a pretest with a small sample of patients who had a stroke. We selected five patients with primary school education and five elderly individuals to participate in this pretest phase. Our research assistants carefully reviewed the questionnaires completed by these participants and identified any questions that were unclear or required further explanation. We then made necessary revisions to ensure that all participants would understand the questions accurately during the formal investigation.

### Measures

#### The demographic questionnaire

The demographic and disease-specific data questionnaire was designed and used to collect information of age, gender, marriage, residential area, educational level, working state, stroke type, family history, number of strokes, monthly income, primary caregiver, sleep conditions and number of chronic diseases.

#### The Stroke Recurrence Risk Perception scale

The Stroke Recurrence Risk Perception Scale (SRRPS) is a tool used to assess stroke patients' recurrence risk perception of their disease. Our research group[16] developed the scale in 2022, which includes 2 sections with 20 items. The first section contains three items that assess the recurrence risk perception of patients with stroke. The second section contains 17 items in three domains: perceived illness risk factors, perceived behavioural risk factors and perceived severity. There are four items in the 'perceived illness risk factors' dimension which includes patients' perception of illness risk factors such as hyperlipidaemia, hyperglycaemia, hypertension and atrial fibrillation of stroke recurrence. There are six items in the 'perceived behavioural risk factors' dimension, which refers to patients' perception of behavioural risk factors such as vegetables, fruits, and salt intake, smoking, and alcohol consumption, and exercise to prevent stroke recurrence. There are seven items in the 'perceived severity' dimension, which means patients' perceptions of harm from recurrence on daily activities, cognition, mood, level of impairment and number of recurrences. Higher scores indicate greater risk awareness, and the score for this section is calculated independently of the overall score. The second section of items ranges from 1 to 3 (disagree, not clear and agree). The Cronbach's alpha of the second section is 0.850 and the Cronbach's alpha for each dimension are 0.875, 0.815 and 0.804. This scale was applied by Tang *et al* in 236 patients with first-episode patients who had ischaemic stroke in China and the Cronbach's alpha for the second part was 0.890.[36] The validity of this scale has also been verified by validation factor analysis in another study.[37] In this study, we used the second section of the scale to examine stroke patients' perception of recurrence risk.

#### The Perceived Social Support Scale

The PSSS is an instrument to measure the individual's self-perception of multidimensional social support. The scale was developed by Zimet in 1987 and revised into a Chinese version by Jiang[38] in 1999. It includes three dimensions of family support, friend support and other support (relatives, leaders, colleagues, etc), with 12 items. Each dimension has four items. The elements in the various dimensions indicate whether the individual requested assistance from others (families, friends, relatives, etc) and whether others were able to assist promptly. It is a 7-point Likert scale (from 1=extremely uncompliant to 7=extremely compliant). The total score of PSSS

ranges from 12 to 84 points, scores ranging from 12 to 36 points indicate low perceived support, 37 to 60 points indicate intermediate perceived support and 61 to 84 points indicate high perceived support. The Cronbach's alpha value of PSSS is 0.92. This scale has been applied to patients who had a stroke in other studies.[39 40]

### The Health Behaviour Scale for Stroke Patients

The HBS-SP is a scale to assess the level of health behaviour in patients with stroke. The scale was developed by Wan et al.[41] It includes 6 dimensions and 25 items. The exercise dimension contains six items, it refers to duration, frequency, type, intensity, plans and motivation of exercise. There are four items in the medication-taking dimension, which means knowledge of taking medication and compliance. The instructions dimension (four items) mainly includes adherence to physician's orders for moderate intake of salt, sugar, oil, etc. The responsibility dimension contains three items, which include paying attention to ingredient lists on food package labels and monitoring heart rate during exercise. The nutrition dimension includes six items, which means eating appropriate amounts of meat, eggs, milk, grains, fruits, soy products, etc. There are two items in the smoking and alcohol dimension, which implies appropriate levels of smoking and alcohol consumption. It is a 4-point Likert scale (from 1=never to 4=routinely performed). But the items of medication and tobacco, and alcohol were reversed. Higher scores indicate better health behaviour. An average score of 2.5 points (between 'sometimes' and 'often') is considered an intermediate level of health behaviour. The Cronbach's alpha value of the HBS-SP scale is 0.878.

### Statistical analysis

The SPSS software V.26 (IBM, Released, 2017) was used to analyse the data. Participants' social demographics characteristics and other related variables were examined by descriptive statistics of frequencies and percentages. Pearson correlation analysis was used to analyse the interrelationships between the three variables. We applied the testing procedure recommended by Wen and Ye[42] and the mediation model was analysed using model 4 in the PROCESS Marco.[43] For the best test of the mediation effect, we carried out the bootstrapping procedure to measure the indirect effect, and a 95% CI was estimated. A p<0.05 was considered statistically significant. A 95% CI excluding 0 was regarded as statistically significant.

### Patient and public involvement

Patients or the public were not involved in the design, conduct, reporting or dissemination of our research.

## RESULTS

### General demographics

A total of 254 questionnaires were obtained. However, four invalid questionnaires were removed because the

**Table 1** Scores on the SRRPS, PSSS and HBS-SP

| Variables | Items | Range of score | Mean | SD |
|---|---|---|---|---|
| SRRPS | 17 | 17–51 | 41.04 | 8.64 |
| Perceived illness risk factors | 4 | 6–18 | 19.96 | 4.76 |
| Perceived behavioural risk factors | 6 | 4–12 | 22.80 | 3.35 |
| Perceived severity | 7 | 7–21 | 17.84 | 4.05 |
| PSSS | 12 | 40–80 | 65.86 | 8.78 |
| Family support | 4 | 13–28 | 23.08 | 3.58 |
| Friend support | 4 | 7–28 | 19.97 | 4.77 |
| Other support | 4 | 12–28 | 22.80 | 3.35 |
| HBS-SP | 25 | 28–81 | 56.78 | 9.03 |
| Exercise | 6 | 5–24 | 11.83 | 4.81 |
| Medication taking | 4 | 5–19 | 11.74 | 2.66 |
| Instructions | 4 | 4–17 | 8.81 | 2.58 |
| Responsibility | 3 | 3–8 | 3.25 | 3.91 |
| Nutrition | 6 | 6–24 | 14.48 | 1.69 |
| Smoking and alcohol | 2 | 2–8 | 6.66 | 0.73 |

HBS-SP, Health Behaviour Scale for stroke patients; PSSS, Perceived Social Support Scale; SRRPS, Stroke Recurrence Risk Perception Scale.

questionnaires with a completion rate of less than 95%. Finally, the number of effective participants was 250, and the final effective rate of questionnaire collection was 98.4%. Over half (70.8%) of patients were elderly patients (≥60 years). Less than half of the patients in the sample had a primary school degree or less (46.1%). Moreover, 34.8% were working and a majority (97.6%) of patients with stroke were ischaemic stroke. In addition, 50.4% of patients who had a stroke were from low-income and middle-income groups. 52% of them were first-episode patients. A comparison of the scales' scores of patients who had a stroke with different characteristics is shown in online supplemental table.

### Descriptive statistics of the SRRPS, PSSS and HBS-SP

Table 1 displays mean scores, SD and ranges of SRRPS, PSSS and HBS-SP. The mean and SD of the SRRPS were 41.04±8.64 scores. The mean and SD of the PSSS were 65.86±8.78 scores. The mean and SD of the PSSS were 56.78±9.03 scores. Therefore, among the patients with stroke, we recruited, recurrence risk perception and PSS were at high levels. Health behaviour was at a moderately low level.

### Correlation analysis between SRRPS, PSSS and HBS-SP

Pearson's correlation analysis was used to investigate the relationships among recurrence risk perception, health behaviour and PSS in patients with stroke. Briefly,

recurrence risk perception was positively correlated with health behaviour (r=0.679, p<0.001), PSS was positively correlated with health behaviour (r=0.778, p<0.001), recurrence risk perception was positively correlated with PSS (r=0.639, p<0.001), meaning that recurrence risk perception, health behaviour and PSS were positively correlated with each other in patients with stroke.

## Mediation analysis of SRRPS, PSSS and HBS-SP

Building on our initial findings from the correlation analysis, we next performed mediation analysis to explore the potential association underlying the relationship between an individual's PSS, their perception of stroke recurrence risk and their adoption of healthy behaviours among patients who had a stroke. We controlled for some general demographic factors (such as age, gender, monthly income and education) which are considered important predictors that affect health behaviour in patients with stroke in China. In model 1, recurrence risk perception had a direct effect on health behaviour (β=0.607, p<0.001). In model 2, recurrence risk perception had a direct effect on PSS (β=0.604, p<0.001). In model 3, recurrence risk perception and PSS positively related to health behaviour (β=0.247, β=0.596, p<0.001), but the comparison of the effect coefficients showed that the effect of recurrence risk perception on health behaviour was diminished from 0.607 to 0.247, which meant that PSS partially mediated the effect between recurrence risk perception and health behaviour, as shown in table 2.

To ensure the accuracy of the test, this study further used the bootstrap method[43] to test the mediating effect of PSS between recurrence risk perception and health behaviour with 5000 replicate samples. The results are shown in table 3, and the 95% CI of the mediating effect of PSS was (0.26 to 0.46), which did not contain 0, indicating that the mediating effect of PSS between recurrence risk perception and health behaviour was established, and the effect accounted for 59.31% (table 3). The mediating model of PSS is constructed in figure 2.

## DISCUSSION

We found that 78.4% of patients with stroke scored less than 2.5 on the HBS, which meant they had moderately low health behaviour. This finding was consistent with previous studies. Li et al[44] investigated 462 patients with stroke in three tertiary hospitals in Liaoning Province, China, and found that patients with stroke had moderate health behaviour. The same results were reaffirmed in another study.[45] More than half of the 231 patients with stroke had moderately low health behaviour. This highlighted the urgent need for further research and focus on the influencing factors of healthy behaviour among patients who had a stroke.

This study demonstrated that PSS was positively correlated with health behaviour. This finding was consistent with our hypothesis 3 and previous studies.[46 47] A cross-sectional study[48] enrolled 133 hypertensive stroke

**Table 2** Mediation results for recurrence risk perception predicting health behaviour mediated by perceived social support

|  | SE | β | t | P value |
|---|---|---|---|---|
| Recurrence risk perception → health behaviour | | | | |
| Age | 0.968 | −1.246 | −1.287 | 0.200 |
| Gender | 0.863 | 0.958 | 1.110 | 0.268 |
| Education | 0.460 | 1.187 | 2.579 | 0.011 |
| Monthly income | 0.742 | 1.158 | 1.561 | 0.120 |
| Recurrence risk perception | 0.053 | 0.607 | 11.529 | 0.001 |
| Recurrence risk perception →perceived social support | | | | |
| Age | 0.997 | 0.240 | 0.240 | 0.810 |
| Gender | 0.890 | 0.912 | 1.026 | 0.306 |
| Education | 0.474 | 1.669 | 3.522 | 0.005 |
| Monthly income | 0.764 | −1.514 | −1.981 | 0.049 |
| Recurrence risk perception | 0.054 | 0.604 | 11.137 | 0.001 |
| Recurrence risk perception, perceived social support → health behaviour | | | | |
| Age | 0.766 | −1.389 | −1.812 | 0.071 |
| Gender | 0.685 | 0.415 | 0.607 | 0.545 |
| Education | 0.373 | 0.193 | 0.516 | 0.606 |
| Monthly income | 0.592 | 2.059 | 3.480 | 0.006 |
| Recurrence risk perception | 0.051 | 0.247 | 4.833 | 0.001 |
| Perceived social support | 0.049 | 0.596 | 12.110 | 0.001 |

patients 6 months after discharge in China have shown the perceptions of chronic illness resources received from healthcare teams, family and friends, and the community were positively correlated with health behaviour. Tan et al[49] conducted a cross-sectional study among 350 stroke participants in China and found that PSS had a direct positive effect on rehabilitation motivation. Moreover, Bandura's social theory[50] also proposed that factors such as social support were actual aspects of the incidence of behaviour. Brouwer-Goossensen et al have found social support was a positive factor in health behaviour change in interviews with 18 patients who had a stroke.[51] Our study provided further evidence of the relationship between PSS and

**Table 3** Mediating model examination by bootstrap

| Recurrence risk perception → health behaviour | | | | |
|---|---|---|---|---|
|  | Effect | SE | LL 95% CI | UL 95% CI |
| Indirect effect | 0.360 | 0.050 | 0.263 | 0.460 |
| Direct effect | 0.247 | 0.512 | 0.147 | 0.348 |

LL, lower limit; UL, upper limit.

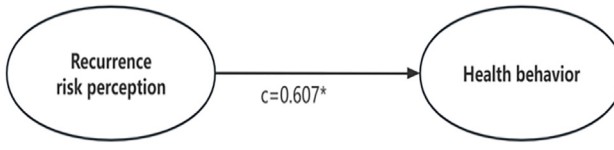

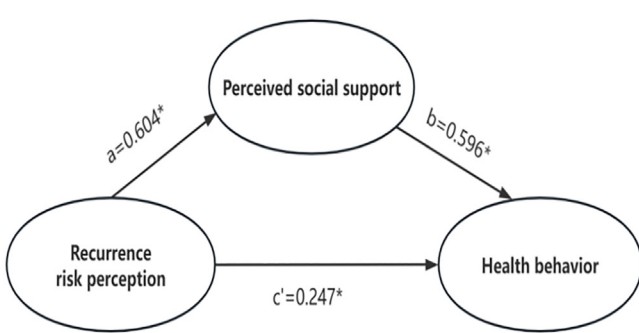

**Figure 2** Mediating effects of perceived social support on the relationship between recurrence risk perception and health behaviour (*p<0.001).

health behaviour. Therefore, PSS is an important factor in improving health behaviour in patients with stroke. In addition, there was a significant positive correlation between recurrence risk perception and PSS. As a result, our initial hypothesis 2 was contradicted by the findings of our study. In other words, PSS positively and significantly related to health behaviour. We speculated the reason for this result was that the higher stroke patients' recurrence risk perception, the more control they had over their diseases. This sense of control over the disease could partly increase their self-efficacy[52] to stay in a good mind, promoting a positive mindset and concurrently enhancing their perception of social support. Calderon et al[53] discovered that a majority of chronic patients expressed a desire for increased information and greater involvement in the decision-making process. To the best of our knowledge, our study was the first study to investigate the association between stroke patients' perceptions of recurrence and their PSS in China.

Most importantly, we found that recurrence risk perception appeared to correlate with health behaviour via two pathways, a direct pathway and an indirect pathway mediated by PSS. (1) The first pathway, recurrence risk perception was correlated with health behaviour significantly and positively. This was contrary to our research hypothesis 1. It may be because patients who had a stroke did not view the risk of recurrence as a frightening event. They gradually increased their confidence in their recovery as they acquired more knowledge about the risk instead. In this way, they were better able to cope with the disease. For example, they began to adopt healthy behaviours to prevent recurrence. These findings were consistent with the HBM.[18] Besides, our previous research found patients with stroke who lacked a perception of the risk of recurrence and preventive strategies tended to limit their

regular hospital visits or refrain from adopting healthy behaviours aimed at preventing recurrence.[40] In other words, recurrence risk perception was an important positive factor of health behaviour. (2) The second pathway was that recurrence risk perception mediated the effect of PSS on health behaviour. This was also contrary to our research hypothesis 4. To effectively implement secondary stroke prevention measures, patients with stroke must have a comprehensive understanding of the risk factors associated with recurrent stroke. Their understanding of the lifestyle changes needed to reduce the chance of future recurrences is equally crucial.[54] In patients with stroke, the perceived risk of recurrence involves not only just concern and fear but also the understanding of early warning signs and risk factors associated with stroke recurrence. So patients with stroke with a higher recurrence risk perception also demonstrated a greater depth of stroke knowledge,[55] and these patients showed a higher level of confidence in recovery than patients with a lower perceived risk of recurrence. Therefore, they perceive information or emotional support from those around them better. The presence of these social supports played a significant role in improving their emotional well-being, consequently fostering a greater willingness to embrace healthier behaviours.

Overall, our study found that recurrence risk perception was not only directly and positively associated with the health behaviours of patients with stroke, but also further related to health behaviour via PSS. This was consistent with the connotation of the CSM model.[26] In other words, our findings provided practical support for the CSM model. As the practical implication is that our finding may aid future research in identifying entry points for health behaviour interventions by taking the methods that improve stroke patients' perception of recurrence risk and PSS. For example, healthcare providers could consider providing patients with recurrence risk communication. Specifically, healthcare professionals can help patients to have an accurate understanding of the risk of recurrence. First, a comparison between the subjective and objective risk perception scores for stroke recurrence is necessary for healthcare professionals. If the stroke recurrence risk perception and objective risk perception scores are inconsistent, then risk communication should be initiated with patients who had a stroke.[56] Common risk communication tools mainly include graphical[57] and numerical forms.[58] To enable patients to correctly grasp their rates of recurrence, appropriate risk communication strategies should be chosen according to their various levels of education and preferences. In addition, healthcare professionals can provide patients who had a stroke with the necessary knowledge and skills to prevent recurrence, such as the aura of stroke recurrence, risk factors of recurrence, and prognosis of recurrence. By doing this, stroke patients' perceptions of social support can be enhanced, which will ultimately help them adopt healthy behaviours. It can also boost their confidence in their ability to recover.

The study still has some limitations. First, it employed a cross-sectional design, thereby precluding the establishment of a causal relationship between recurrence risk perception and health behaviour. Nonetheless, the study successfully confirmed a significant correlation between recurrence risk perception and health behaviour. Second, despite our efforts to incorporate multicentre sampling, the impact of COVID-19 necessitated the restriction of our study to a single tertiary hospital in Henan Province. Therefore, the findings may not be generalisable to all patients with stroke. Besides, only six haemorrhagic patients with stroke were included in our study, so it was difficult for us to explore differences in stroke type in patients' perceived risk of recurrence, PSS and health behaviour. Lastly, our study included a relatively elderly population of patients with stroke, which may have some impact on the generalisability of the findings. Therefore, we recommend the design of a multicentre longitudinal study specifically focusing on young patients who had a stroke in future research. Such a study design would help clarify the dynamic trends in recurrence risk perception, PSS and health behaviour over time.

## CONCLUSION

In conclusion, this study found that PSS mediated the relationship between recurrence risk perception and health behaviour. Our findings expand on previous research on the interplay between recurrence risk perception, PSS and health behaviour in patients with stroke. The results of this study can serve as a useful guide for future clinical nursing practice and research in this area. Considering the importance of recurrence risk perception and PSS, it is recommended that these factors be incorporated into the assessment process for patients with stroke, both during hospitalisation and follow-up care. We suggest that healthcare professionals should assist patients with stroke in developing an accurate perception of recurrence risk with appropriate recurrence risk communication approaches. This method can significantly improve patients' capacity to identify and access social support, ultimately enhancing their confidence in recovery and motivation to engage in healthy behaviours. Moreover, it is crucial to recognise the significance of social support from family and friends as a primary source of PSS. Healthcare professionals should encourage and educate patients' loved ones to offer emotional support, which can positively impact patients' recovery and well-being.

**Acknowledgements** We would like to acknowledge the tertiary hospital of Henan province for their cooperation. We greatly appreciate the support of all patients for their voluntary participation in this study.

**Contributors** Z-XZ and B-LL designed the study. XW, HJ and WW contributed to writing and revising. XW, Y-XM and CZ were in charge of data collection and analysis. QZ and S-YC were in charge of study design and essential help. All authors read and approved the final manuscript. Finally, Z-XZ accepts full responsibility for the work and/or the conduct of the study as guarantor, has access to the data, and controlled the decision to publish.

**Funding** This work was supported by The National Natural Science Foundation of China Youth Science Foundation Project (72104221).

**Competing interests** None declared.

**Patient and public involvement** Patients and/or the public were not involved in the design, or conduct, or reporting, or dissemination plans of this research.

**Patient consent for publication** Not applicable.

**Ethics approval** This study involves human participants and the Research Ethics Committee of Zhengzhou University (ZZURIB2020-08) approved the study. Participants gave informed consent to participate in the study before taking part.

**Provenance and peer review** Not commissioned; externally peer reviewed.

**Data availability statement** Data are available upon reasonable request.

**ORCID iDs**
Xiaoxuan Wang http://orcid.org/0000-0002-8549-7698
Hu Jiang http://orcid.org/0000-0001-6301-0829
Yong-Xia Mei http://orcid.org/0000-0003-4269-3231

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
