## [Reviewer comments · BMJ Open]

ARTICLE DETAILS

TITLE (PROVISIONAL)	Mediation role of perceived social support between recurrence risk perception and health behavior among patients with stroke in China: a cross-sectional study
AUTHORS	Wang, Xiaoxuan; Zhang, Zhen-xiang; LIN, Bei-lei; Jiang, Hu; Wang, Wenna; MEI, YONG-XIA; Zhang, Chunhui; Zhang, Qiushi; CHEN, Su-yan

VERSION 1 – REVIEW

REVIEWER	Rezarta Lalo Universiteti Ismail Qemali Vlore, Health care
REVIEW RETURNED	13-Oct-2023

GENERAL COMMENTS	The introduction section is very long and looks more like a literature review study than a summary that identifies the problem. You have described more global than local statistics of this phenomenon in China or Luoyang City, Henan Province. There are many definitions of terms in the introduction section which makes it tedious for the reader. Specifically, you describe unnecessary data related to the concept of recurrence risk perception. On the one hand, describe what this term includes, and on the other hand, you say that there is no exact definition. Then, why is it important to evaluate it, when you give different versions of this concept. The aim of the study is not related to how different authors define this term, but to the fact why it is important to measure it and what are the results in similar studies. I would suggest that you mention the importance of measuring or evaluating this factor. I would also suggest that in the introduction section you should provide more information on the theoretical framework of the study and the importance of the CSM model, its validity in other studies in order to have more references of its use as well as to explained the reasons why this study is based on this theoretical framework. You also mention that limited research exists on the impact of recurrence risk perception on perceived social support and its associated psychological effects among stroke patients (line 86). Where do you base this conclusion? Could you provide a reference that supports this statement? In the methodology section, the point b of the inclusion criteria is not clear. In order for the patient to have the ability to communicate, he must be in a stable stage, not in an acute stage. Explain better what stable stage means. In line 132 you mention that the SRRPS scale was applied to the
--

	Chinese population. Are there scientific references or studies that have tested the validity of this scale? Could you mention some studies that have used this questionnaire as a measurement tool? It is not clear what are the items for each domains of the questionnaire. I would also suggest a supplementary file that provides numerical and percentages results for each items. Table 1 of the results section presents only the results of the overall mean scores for each scale, but does not present the mean scores for each domains of the 3 questionnaires. I would suggest a table with descriptive and analytical results, for example, what percentage of participants have family support, etc. No clear and concise results are given for each domain, for example what is the mean score or percentage of patients regarding perceived severity or behavior. There is also no precise data on health-related behaviors such as nutrition, smoking, alcohol, etc Tables have analytical data or describe statistical relationships rather than descriptive data. I would suggest a brief description of the results of Table 1 as a clearer information for the reader. The discussion section begins as a literature review by repeating information described in the introduction section. Please avoid the repetitions! Lines 234–236. Line 246/247 is almost similar to line 53/54 of the introduction section. I would suggest that the discussion section analyzes the results of the current study you have undertaken by comparing them with the results of other similar studies. Line 235/238 is an unnecessary paragraph in the discussion section, but can be mentioned in the introduction section to explain the reason for analyzing these factors. You mentioned on line 260 of the discussion section that our initial hypothesis was contradicted by the findings of our study. What is the initial hypothesis of your study? Please describe clearly what is the hypothesis or research question of the current study.
--	---

REVIEWER	L Guo The First Affiliated Hospital of Zhengzhou University
REVIEW RETURNED	14-Oct-2023

GENERAL COMMENTS	Overall, I found this paper an interesting evaluation of the relationship between recurrence risk perception and health behavior with stroke patients. The suggestions are as follows. Introduction The introduction paragraph overemphasizes the current state of unhealthy behaviors, perhaps shortening this might be useful. There is a lack of description of the relationship between recurrence risk perception and health behavior, and it is recommended to supplement it. Recurrence risk perception is not yet defined in the introduction section. Health behavior is influenced by multiple factors, and the introduction paragraph fails to adequately elucidate the significance of recurrence risk perception in relation to health behavior. It is advisable to consider reorganizing the preface section. Methods Participants and procedures Is there a difference in the perceived risk of recurrence between
---

	ischemic and hemorrhagic patients. Will the inclusion of the population affect the accuracy of the results. please, precise participants. Results The authors stated, "Over half (70.8%) of patients were elderly 182 patients (≥60 years)". Did it have an impact on the validity of research findings. Discussion I wonder if there is a positive correlation between the perception of recurrence risk and health behavior. There seems to be no explanation in the discussion. How to intervene in the perception of recurrence risk in stroke patients in the future, how to inform patients of the risk of recurrence, and how to guide clinical application. It is necessary for authors to clarify them in the discussion.
--	---

VERSION 1 – AUTHOR RESPONSE

Response to reviewer:

Thanks for your suggestions, we revised this section as follows:

Mediation role of perceived social support between recurrence risk perception and health behavior among patients with stroke in China: a cross-sectional study

2. In general, please work to improve the quality of the writing throughout your manuscript. We recommend asking a colleague who is proficient in written English to assist you; alternatively, you could enlist the help of a professional copy editing service.

Response to reviewer:

Thanks for your good advice. We invited our English major colleagues to help us revise the manuscript, and after the revision was completed, we conducted further review to avoid any change in the meaning of the sentences we wanted to elaborate.

3. Abstract >> objectives: "to explore whether perceived social support would function as mediators" Should this be: "to explore whether perceived social support would function as a mediator"?

Response to reviewer:

Please forgive us for misusing the "to explore whether perceived social support would function as a mediator" instead of "to explore whether perceived social support would function as mediators", which we have corrected in the manuscript.

4. Can you add the relevant numbers/ statistics to support the following statement in the results section of the abstract?: "Perceived social support mediated the relationship between recurrence risk perception and health behavior, after controlling for age, sex, education, monthly income in the regression models."

Response to reviewer:

Thanks for your suggestions, we revised this section as follows:

Perceived social support mediated the relationship between recurrence risk perception and health behavior, after controlling for age, gender, education, monthly income in the regression models (95% CI=[0.263,0.460]).

5.- Abstract: "And perceived social support had.." Please do not start a new sentence with 'And'.

Response to reviewer:

Thanks for your suggestions, we revised this section as follows:

It also confirmed that perceived social support had a highest mediation effect with a proportion of mediation up to 59.31%.

6. Please revise the first bullet point of the 'Strengths and limitations' section of your manuscript (after the abstract). This section should contain up to five short bullet points, no longer than one sentence each, that relate specifically to the methods. The novelty, aims, results or expected impact of the study should not be summarised here.

Response to reviewer:

Thanks for your suggestions, we revised this section as follows:

STRENGTHS AND LIMITATIONS OF THIS STUDY

- A path analysis is used to analyse the associations among perceived social support, recurrence risk perception and health behavior.
- This study is a cross-sectional survey and therefore cannot make causal inferences.
- Survey data from only one region may limit the generalizability of our findings.

3. The manuscript contains redacted information (e.g. the name of the ethics committee that approved the study). Please note that BMJ Open uses fully open peer review so you do not need to conceal this information. Please can you add this information back into the paper?

Response to reviewer:

Thanks for your good advice. We have added the information back into the paper.

Reply to Reviewer 1

1. The introduction section is very long and looks more like a literature review study than a summary that identifies the problem. You have described more global than local statistics of this phenomenon in China or Luoyang City, Henan Province.

Response to reviewer:

Thank you for your detailed review, which has helped us tremendously in improving the quality of our articles. We have partially trimmed the introduction section of the article and replaced the partly global statistics with local statistics of this phenomenon.

2. There are many definitions of terms in the introduction section which makes it tedious for the reader. Specifically, you describe unnecessary data related to the concept of recurrence risk perception. On the one hand, describe what this term includes, and on the other hand, you say that there is no exact definition. Then, why is it important to evaluate it, when you give different versions of this concept. The aim of the study is not related to how different authors define this term, but to the fact why it is important to measure it and what are the results in similar studies.

I would suggest that you mention the importance of measuring or evaluating this factor.

Response to reviewer:

Thanks for your good advice. We have deleted the unnecessary introduction of concepts and mentioned the importance of measuring or evaluating recurrence risk perception. We revised this section as follows:

The study concluded that stroke patients' accurate perception of recurrence risk was an important predictor for patients to adopt healthy behaviors to control and reduce the recurrence of stroke [17]. However, the precise relationship between risk perception of recurrence and health behavior actually remains uncertain.

3. I would also suggest that in the introduction section you should provide more information on the theoretical framework of the study and the importance of the CSM model, its validity in other studies in order to have more references of its use as well as to explained the reasons why this study is based on this theoretical framework.

Response to reviewer:

Thanks for your good advice. We added relevant statements about the CSM model, specifically the current application of the model in other studies and the reasons why the model serves as the conceptual framework for our study. The added sentences are as follows:

The Common Sense Model of self-regulation (CSM) is a theoretical framework which is widely used in the study of illness perception [25]. It was designed to describe dynamic interactions among the variables controlling health behavior in response to current or future health threats [26]. The CSM comprises six main components [29]: (1) situational stimuli, (2) cognitive illness representations, (3) emotional illness representations, (4) coping strategies, (5) illness and emotional outcomes, and (6) coping appraisal. Previous studies have already applied the CSM model to predict health outcomes in diabetes [27], explore the experience of breathlessness [28] and predict healthy eating among individuals at risk of metabolic syndrome [29]. For patients with stroke, the surrounding situational stimuli (including endogenous or social factors) cause patients with stroke to develop cognitive and emotional illness representation (recurrence risk perception). Cognitive and emotional illness representation are also continually updated [29] in response to memories of new information (e.g., perceived social support), which affects coping strategies (the choice of adopting health behavior). Therefore, it is reasonable to choose the CSM model as the theoretical framework for our study.

4. You also mention that limited research exists on the impact of recurrence risk perception on perceived social support and its associated psychological effects among stroke patients (line 86). Where do you base this conclusion? Could you provide a reference that supports this statement?

Response to reviewer:

Thank you for pointing out this problem. Some studies have found that social support had an effect on recurrence risk perception, but the effect of recurrence risk perception on perceived social support still needs to be explored. We have added the reference in the manuscript.

5. In the methodology section, the point b of the inclusion criteria is not clear. In order for the patient to have the ability to communicate, he must be in a stable stage, not in an acute stage. Explain better what stable stage means.

Response to reviewer:

We apologize for the inappropriate narrative. We have amended it in the manuscript. In fact, when we recruited patients at recovery stage who had good communication skills, and we have now redefined such patients.

(b) at recovery stage (2 weeks to 6 months after onset) [34]

6. In line 132 you mention that the SRRPS scale was applied to the Chinese population. Are there scientific references or studies that have tested the validity of this scale? Could you mention some studies that have used this questionnaire as a measurement tool?

Response to reviewer:

Thank you for pointing out this problem. We have added it in the manuscript. The added sentences are as follows:

This scale was applied by Tang in 236 patients with first-episode ischemic stroke patients in China and the Cronbach's alpha for the second part was 0.890 [36]. The validity of this scale has also been verified by validation factor analysis in another study [37].

7. It is not clear what are the items for each domains of the questionnaire. I would also suggest a supplementary file that provides numerical and percentages results for each items.

Response to reviewer:

Thank you for pointing out this problem. We apologize that we did not describe clearly the items for each dimension of the scale. We have added it in the measures and results of manuscript. The added sentences are as follows:

It includes three dimensions of family support, friend support, and other support (relatives, leaders, colleagues, etc.), with 12 items. Each dimension has 4 items.

It includes 6 dimensions of exercise(6 items), medication taking(4 items), instructions(4 items), responsibility(3 items), nutrition(6 items), and smoking and alcohol(2 items), with 25 items.

In addition, if you are interested in the stroke recurrence risk perception scale, we will send you the scale, but we do not wish to publicly present it in the supplemental file.

8. Table 1 of the results section presents only the results of the overall mean scores for each scale, but does not present the mean scores for each domains of the 3 questionnaires.

Response to reviewer:

Thank you for pointing out this problem. We have added it in the Table 1.

Table 1 Scores on the SRRPS, PSSS and HBS-SP

Variables	Items	Range of score	Mean	Stand deviation
SRRPS	17	17~51	41.04	8.64
Perceived illness risk factors	4	6~18	19.96	4.76
Perceived behavioral risk factors	6	4~12	22.80	3.35
Perceived severity	7	7~21	17.84	4.05
PSSS	12	40~80	65.86	8.78
Family support	4	13~28	23.08	3.58
Friend support	4	7~28	19.97	4.77
Other support	4	12~28	22.80	3.35
HBS-SP	25	28~81	56.78	9.03
Exercise	6	5~24	11.83	4.81
Medication taking	4	5~19	11.74	2.66
Instructions	4	4~17	8.81	2.58
Responsibility	3	3~8	3.25	3.91
Nutrition	6	6~24	14.48	1.69
Smoking and alcohol	2	2~8	6.66	0.73

Abbreviations: SRRPS, stroke recurrence risk perception scale; PSSS, Perceived social support scale; HBS-SP, Health behavior scale for stroke patients

9. I would suggest a table with descriptive and analytical results, for example, what percentage of participants have family support, etc. No clear and concise results are given for each domain, for example what is the mean score or percentage of patients regarding perceived severity or behavior.

There is also no precise data on health-related behaviors such as nutrition, smoking, alcohol, etc

Response to reviewer:

Thank you for your good advice. We have amended it in the supplemental file.

10. Tables have analytical data or describe statistical relationships rather than descriptive data. I would suggest a brief description of the results of Table 1 as a clearer information for the reader.

Response to reviewer:

Thank you for your good advice. We have added it in the manuscript. The added sentences are as follows:

The mean and SD of stroke recurrence risk perception scale were 41.04 ± 8.64 scores. The mean and SD of perceived social support scale were 65.86 ± 8.78 scores. The mean and SD of perceived social support scale were 56.78 ± 9.03 scores. Therefore, among the patients with stroke we recruited, recurrence risk perception and perceived social support were at high levels. Health behavior was at a moderately low level.

11. The discussion section begins as a literature review by repeating information described in the introduction section. Please avoid the repetitions! Lines 234–236. Line 246/247 is almost similar to line 53/54 of the introduction section.

Response to reviewer:

Thank you for your good advice. We removed excessive and unnecessary literature reviews and added comparisons of the results of our study with similar studies. The added sentences are as follows:

We found that 78.4% of patients with stroke scored less than 2.5 on the health behavior scale, which meant they had moderately low health behavior. This finding was consistent with previous studies. Li [42] investigated 462 patients with stroke in three tertiary hospitals in Liaoning Province, China, and found that patients with stroke had moderate health behavior. The same results were reaffirmed in Liu's study [43]. More than half of the 231 patients with stroke had moderately low health behavior. This highlighted the urgent need for further research and focus on the influencing factors of healthy behaviors among stroke patients.

12. I would suggest that the discussion section analyzes the results of the current study you have undertaken by comparing them with the results of other similar studies.

Response to reviewer:

Thank you for your good advice. We have amended it in the manuscript. The added sentences are as follows:

We found that 78.4% of patients with stroke scored less than 2.5 on the health behavior scale, which meant they had moderately low health behavior. This finding was consistent with previous studies. Li et al [41] investigated 462 patients with stroke in three tertiary hospitals in Liaoning Province, China, and found that patients with stroke had moderate health behavior. The same results were reaffirmed in another study [42]. More than half of the 231 patients with stroke had moderately low health behavior.

This study demonstrated that perceived social support was positively correlated with health behavior. This finding was consistent with past studies [43,44]. A cross-sectional study [43] based on data from the Spanish National Health Survey (ENSE2017) with a final sample of 1006 individuals with diabetes mellitus has shown the perceived social support was positively correlated with physical activity. The same results were confirmed in another study. Guo et al [44] conducted a community-based cross-sectional study among 1697 participants with hypertension in China and found social support had a direct positively effect on medication adherence.

13. Line 235/238 is an unnecessary paragraph in the discussion section, but can be mentioned in the introduction section to explain the reason for analyzing these factors.

Response to reviewer:

Thank you for your good advice. We have moved line 235/238 to the introduction section.

14. You mentioned on line 260 of the discussion section that our initial hypothesis was contradicted by the findings of our study. What is the initial hypothesis of your study? Please describe clearly what is the hypothesis or research question of the current study.

Response to reviewer:

Thank you for pointing out this problem. We did not narrate the research hypothesis clearly. We have added the speculative process of hypothesis to the introduction. Moreover, we have listed the hypothesis section separately. The added sentences are as follows:

Hypothesis 1 – Recurrence risk perception affects health behavior directly and negatively.

Hypothesis 2 – Recurrence risk perception affects the perceived social support directly and negatively.

Hypothesis 3 – Perceived social support affects health behavior directly and positively.

Hypothesis 4– Recurrence risk perception indirectly affects health behavior via Perceived social support.

Reply to Reviewer 2

1. Introduction

The introduction paragraph overemphasizes the current state of unhealthy behaviors, perhaps shortening this might be useful.

Response to reviewer:

Thank you for your good advice. We have amended it in the manuscript.

2. There is a lack of description of the relationship between recurrence risk perception and health behavior, and it is recommended to supplement it. Health behavior is influenced by multiple factors, and the introduction paragraph fails to adequately elucidate the significance of recurrence risk perception in relation to health behavior. It is advisable to consider reorganizing the preface section.

Response to reviewer:

Thank you for your detailed review, which has helped us tremendously in improving the quality of our articles. In our previous manuscript, the presentation of studies related to recurrence risk perception and health behavior was not clear enough, so we describe in further detail how previous studies have explored the relationship between recurrence risk perception and health behavior and the findings. The added sentences are as follows:

On the contrary, a cross-sectional comparative study conducted in China showed that patients with stroke who perceived higher risk of recurrence were more likely to adopt healthy behavior [14]. Iversen et al [19] investigated 435 patients with stroke and confirmed that higher seriousness perception of stroke was associated with more help-seeking behavior and shorter prehospital delay. There is also another argument that recurrence risk perception is detrimental to health behavior. Freeman-Gibb et al [20] recruited 117 patients and found patients who perceive a high risk of recurrence always experience higher levels of anxiety and fear. Excessive anxiety and fear were actually detrimental to the patient's adoption of health behavior [21].

3. Recurrence risk perception is not yet defined in the introduction section.

Response to reviewer:

Thank you for pointing out this problem. Our original manuscript introduced the concept of recurrence risk perception as proposed by different researchers, which may have caused confusion. We have revised the manuscript, the added sentence is as follows:

Recurrence risk perception refers to the perception of early warning features of recurrence, severity, behavior-related risk factors, and disease-related risk factors [16].

4. Methods-Participants and procedures

Is there a difference in the perceived risk of recurrence between ischemic and hemorrhagic patients. Will the inclusion of the population affect the accuracy of the results. please, precise participants.

Response to reviewer:

Thank you for pointing out this problem. We reviewed some of the literature and found studies showing less variability in the perceived risk of recurrence in ischemic and hemorrhagic patients. In addition, our research group did not define a certain stroke type as the applicable population when we developed the stroke recurrence risk perception scale, so we did not limit the stroke type when recruiting the population. But we think this may indeed be an influencing factor, which we will add to the limitations and anticipate comparing the differences in the recurrence risk perception between the two types of patients in our next article. We have added the limitation to the manuscript, the added sentence is as follows.

Besides, only six hemorrhagic patients with stroke were included in our study, so it was difficult for us to explore differences in stroke type in patients' perceived risk of recurrence, perceived social support and health behavior.

5. Results

The authors stated, "Over half (70.8%) of patients were elderly 182 patients (≥ 60 years)". Did it have an impact on the validity of research findings.

Response to reviewer:

Thank you for pointing out this problem. More than half of the patients included in our study were elderly stroke patients. This is a limitation of this study. We have added the limitation to the manuscript, the added sentence is as follows:

Lastly, our study included a relatively elderly population of patients with stroke, which may have some impact on the generalizability of the findings.

6. I wonder if there is a positive correlation between the perception of recurrence risk and health behavior. There seems to be no explanation in the discussion.

Response to reviewer:

Thank you for pointing out this problem. We checked the original manuscript and realized that the past representations may have been a bit vague. We further modified the formulation of the relationship between perception of recurrence risk and health behavior to make this content seem clearer. The added sentences are as follows:

First pathway, recurrence risk perception was correlated with health behavior significantly and positively. This was contrary to our research hypothesis. It may be due to the fact that stroke patients did not view the risk of recurrence as a frightening event. They gradually increased their confidence in their own recovery as they acquired more knowledge about the risk instead. In this way, they were better able to cope with the disease. For example, they began to adopt healthy behaviors to prevent recurrence. In fact, this findings were consistent with the health belief model (HBM) [18]. Besides, our previous research found patients with stroke who lacked a perception of the risk of recurrence and preventive strategies exhibited a tendency to limit their regular hospital visits or refrain from adopting healthy behaviors aimed at preventing recurrence [40]. In other words, recurrence risk perception was an important positive factor of health behavior.

7.How to intervene in the perception of recurrence risk in stroke patients in the future, how to inform patients of the risk of recurrence, and how to guide clinical application.
It is necessary for authors to clarify them in the discussion.

Response to reviewer:

Thank you for pointing out this problem. In our previous manuscript, this was partially described but not clearly enough, so we describe it in further detail, including suggestions on the approach to risk communication with stroke patients and the focus of risk communication. The added sentences are as follows:

First, a comparison between the subjective and objective risk perception scores for stroke recurrence is necessary for healthcare professionals. If the stroke recurrence risk perception and objective risk perception scores are inconsistent, then risk communication should be initiated with stroke patients [51]. Common risk communication tools mainly include graphical [52] and numerical forms [53]. To enable patients correctly grasp their rates of recurrence, appropriate risk communication strategies should be chosen according to their various level of education and preferences. In addition, healthcare professionals can provide stroke patients with the necessary knowledge and skills to prevent recurrence, such as the aura of stroke recurrence, risk factors of recurrence, and prognosis of recurrence.

VERSION 2 – REVIEW

REVIEWER	Rezarta Lalo Universiteti Ismail Qemali Vlore, Health care
REVIEW RETURNED	15-Nov-2023

GENERAL COMMENTS	1. I think it is not important to mention only the number of items but a brief information about their content. for example, it is unclear when you say instructions(4 items), responsibility(3 items). What does the instruction or responsibility contain? I suggest that you add some information about the items mentioned or included in the questionnaire. For example, it includes 6 dimensions of exercise, which are related to..... Yes I am interested in the stroke recurrence risk perception scale. Could you send me? 2. I also suggest that you should mention in the discussion of the results whether your results are consistent with or contradict the hypotheses raised. 3. You have mentioned references 43/44 to show that perceived social support was positively correlated with health behavior, but in these studies we do not have populations with patients suffering from stroke. I suggest that you find references to other studies that show this relationship in stroke patients.
--

REVIEWER	L Guo The First Affiliated Hospital of Zhengzhou University
REVIEW RETURNED	17-Nov-2023

GENERAL COMMENTS	The problem I raised before has been properly resolved. But social support includes both subjective and objective aspects, and it seems unclear why PSSS is chosen. It is advisable to supplement relevant
--

VERSION 2 – AUTHOR RESPONSE

Reply to Reviewer 1

1. I think it is not important to mention only the number of items but a brief information about their content. for example, it is unclear when you say instructions(4 items), responsibility(3 items).

What does the instruction or responsibility contain?

I suggest that you add some information about the items mentioned or included in the questionnaire.

For example, it includes 6 dimensions of exercise, which are related to.....

Yes I am interested in the stroke recurrence risk perception scale. Could you send me?

Response to reviewer:

Thanks for your suggestions. We have added the detailed information about the items mentioned or included in the three questionnaires. The section we have revised and the contents of the stroke recurrence risk perception scale are as follows:

There are four items in the “perceived illness risk factors” dimension which includes patients’ perception of illness risk factors such as hyperlipidemia, hyperglycemia, hypertension, and atrial fibrillation of stroke recurrence. There are six items in the “perceived behavioral risk factors” dimension, which refers to patients’ perception of behavioral risk factors such as vegetables, fruits, and salt intake, smoking and alcohol consumption, and exercise to prevent stroke recurrence. There are seven items in the “perceived severity” dimension, which means patients’ perceptions of harm from recurrence on daily activities, cognition, mood, level of impairment, and number of recurrences.

It includes three dimensions of family support, friend support, and other support (relatives, leaders, colleagues, etc.), with 12 items. Each dimension has 4 items. The elements in the various dimensions indicate whether the individual requested assistance from others (families, friends, relatives, etc.) and whether others were able to assist in a timely manner.

The exercise dimension contains six items, it refers to duration, frequency, type, intensity, plans, and motivation of exercise. There are 4 items in the medication taking dimension, which means knowledge of taking medication and compliance. The instructions dimension (4 items) mainly includes adherence to physician’s orders for moderate intake of salt, sugar, oil, etc. The responsibility dimension contains three items, which includes paying attention to ingredient lists on food package labels and monitoring heart rate during exercise. The nutrition dimension includes six items, which means eating appropriate amounts of meat, eggs, milk, grains, fruits, soy products, etc. There are 2 items in the smoking and alcohol dimension, which implies appropriate levels of smoking and alcohol consumption.

The stroke recurrence risk perception scale (section 2)

Please tick (√) your answers in the table based on your current conditions.

Items	Disagree	Not clear	Agree
1. Recurrence can lead to a decrease in self-care in daily life.			
2. Recurrence increases the likelihood and severity of disability.			
3. Recurrence can lead to reduced capacity/scope for social participation.			
4. Recurrence increases the risk of multiple recurrence.			
5. Recurrence increases risk of death.			
6. Relapse increases the incidence of cognitive dysfunction (comprehension, memory, and numeracy etc.)			
7. Relapse increases the incidence of depression.			
8. Inadequate daily fruit intake increases risk of stroke recurrence.			
9. Inadequate daily vegetable intake increases risk of stroke recurrence.			
10. Smoking/secondhand smoke increases risk of stroke recurrence.			
11. Excessive alcohol consumption increases risk of stroke recurrence.			
12. Lack of exercise increases risk of stroke recurrence.			
13. High-salt diet increases risk of stroke recurrence.			
14. High blood cholesterol increases risk of stroke recurrence.			
15. High blood sugar increases risk of stroke recurrence.			
16. High blood pressure increases risk of stroke recurrence.			
17. Atrial fibrillation increases risk of stroke recurrence.			

2. I also suggest that you should mention in the discussion of the results whether your results are consistent with or contradict the hypotheses raised.

Response to reviewer:

Thanks for your good advice. We have added the descriptions in the discussion. We revised this section as follows:

This study demonstrated that perceived social support was positively correlated with health behavior. This finding was consistent with our hypothesis 3 and previous studies [46,47].

As a result, our initial hypothesis 2 was contradicted by the findings of our study.

First pathway, recurrence risk perception was correlated with health behavior significantly and positively. This was contrary to our research hypothesis 1.

The second pathway was that recurrence risk perception mediated the effect of perceived social support on health behavior. This was also contrary to our research hypothesis 4.

3. You have mentioned references 43/44 to show that perceived social support was positively correlated with health behavior, but in these studies we do not have populations with patients suffering from stroke.

I suggest that you find references to other studies that show this relationship in stroke patients.

Response to reviewer:

Thanks for your good advice. We have found the relationship between perceived social support and health behavior which confirmed in the stroke population and replaced the original references 43/44. We have amended it as follows:

A cross-sectional study [48] enrolled 133 hypertensive stroke patients at 6 months after discharge in China has shown the perceptions of chronic illness resources which receive from health care team, family and friends, and community were positively correlated with health

behavior. Tan [49] conducted a cross-sectional study among 350 stroke participants in China and found perceived social support had a direct positively effect on rehabilitation motivation.

Reply to Reviewer 2

1.The problem I raised before has been properly resolved. But social support includes both subjective and objective aspects, and it seems unclear why PSSS is chosen. It is advisable to supplement relevant content in the introduction section.

Response to reviewer:

Thanks for your good advice. Our initial explanation was a bit unclear, and we have amended the part in the introduction section again. We have also added the application of the PSSS scale in other studies in the Measures section. The specific modifications are as follows:

Social support is another significant impact factor of health behavior [22]. Social support includes both subjective and objective aspects, and the subjective aspect refers to perception of social support. It is known that social support is an interactive process that involves providing and receiving social support, which means social support must be perceived by patients so that it can be effective [23]. Besides, a systematic study [24] discovered that the associations between perceived social support and health-related quality of life appear to be more frequently significant than the relationships between specific categories or sources of social support and health-related quality of life.

The total score of PSSS ranges from 12-84 points, scores ranging from 12 to 36 points indicate low perceived support, 37 to 60 points indicate intermediate perceived support, and 61 to 84 points indicate high perceived support. The Cronbach's alpha value of PSSS is 0.92. This scale has been applied to stroke patients in other studies [39,40].

VERSION 3 – REVIEW

REVIEWER	Rezarta Lalo Universiteti Ismail Qemali Vlore, Health care
REVIEW RETURNED	17-Dec-2023
GENERAL COMMENTS	Thank you for responding correctly to my suggestions as a reviewer
REVIEWER	L Guo The First Affiliated Hospital of Zhengzhou University
REVIEW RETURNED	06-Dec-2023
GENERAL COMMENTS	The issue I raised has been properly resolved and I agree to publish it